# Autism Spectrum Disorder Diagnostic Criteria Changes and Impacts on the Diagnostic Scales-Utility of the 2nd and 3rd Versions of the Gilliam Autism Rating Scale (GARS)

**DOI:** 10.3390/brainsci12050537

**Published:** 2022-04-23

**Authors:** Sayyed Ali Samadi, Cemal A. Biçak, Hana Noori, Barez Abdalla, Amir Abdullah, Lizan Ahmed

**Affiliations:** 1Institute of Nursing and Health Research, Ulster University, Newtownabbey BT37 0QB, Northern Ireland, UK; 2Bahoz Centre for Children with Developmental Disabilities, Erbil 44002, The Kurdistan Region of Iraq, Iraq; cemal.a.bicak@bahozcenter.com (C.A.B.); hanan@bahozcenter.com (H.N.); barez@bahozcenter.com (B.A.); amir@bahozcenter.com (A.A.); lezan@bahozcenter.com (L.A.)

**Keywords:** autism, autism spectrum disorder, GARS-2, assessment, GARS-3, screening, low and middle-income countries, Kurdistan

## Abstract

There is joint agreement among professionals internationally on the importance of diagnosing autism spectrum disorders (ASD) in the early stages of the emergence of symptoms. Criteria changes for the diagnosis of ASD need updated versions of the scale to make the diagnosis feasible. This study aimed to evaluate the level of overlap between two different versions of the Gilliam Autism Rating Scale (GARS-2 and GARS-3), which have been updated based on changes in DSM-IV and DSM-5 on a Kurdish sample of individuals at risk of having ASD and Intellectual Disability, referred to the Bahoz center in the Kurdistan Region of Iraq. A group of 148 cases with ASD and developmental disabilities (DD) was evaluated using the 2nd and 3rd versions of the GARS scale to understand the level of cases that confirm an ASD diagnosis in both scales. Ninety-six individuals (65%) scored about the cut-off score for being diagnosed with ASD based on the GARS-2, and 137 individuals (93%) scored above the cut-off score based on the GARS-3. Moreover, keeping updated and meeting the changing demand of standardization and cultural suitability of the updating scales is a challenge. This challenge is due to the shortage of infrastructure sources and lack of established professionals in low- and middle-income countries (LMICs). Findings indicated that GARS-3, updated based on the DSM-5, tends to diagnose children with accompanying diagnoses and different levels of symptoms severity of ASD at different age levels. Further studies are needed to help professionals and policymakers in low- and middle-income countries understand the updated versions of the available scales and depend on the older version, which must be considered cautiously.

## 1. Introduction

Autism spectrum disorder (ASD) is diagnosed following the same criteria, and it is common in every society [1,2,3]. A wealth of findings indicate that if ASD is diagnosed earlier, the individual will benefit from earlier services and specific treatments. Diagnosis at the early stages may eventually improve the symptoms and advance the final intervention outcomes [4,5,6]. ASD diagnosis is not an easy task [7], especially in individuals with accompanying diagnosis, and should be based on a diagnostic process involving extensive information observed from the child, caregivers, and other sources such as educational and training systems. Devices such as observation, interviews, and questionnaires are valuable ways of data collection to make the final diagnosis more reliable.

In 2013, DSM-5, the final version of the diagnostic reference, was published [1]. ASD diagnostic criteria changed, and the most critical change was merging two final versions of DSM [8,9] domains into one, or changing the triad of impairments in ASD to a dyad (integrating two independent social communication and social interaction into one domain). The updated version also requires two indications of the behavioral domain of restricted, repetitive patterns of behavior, interests, or activities, along with the addition of sensory input. In the previous version of the book, the presence of one criterion was required, and the company of sensory issues was not considered. Like DSM-5, Autism in the ICD-11 falls into two categories, and the third category for language problems is excluded. The most important benefit of these changes is that the five disconnected disorders under one umbrella of PDDs were merged to shape one single category of autism spectrum disorder [10].

The conceptual shifts mean that ways of screening and assessing children have to change. Early tools and measures may no longer be suitable and may fail to identify children who have the condition and, hence, lose out on receiving assistance. Therefore, updating the available screening and diagnosing scales is a necessary step to take. Presently, a common practice among clinicians and researchers globally is to rely on different instruments for screening, diagnosis, and assessments of ASD, including identifying individual characteristics, intervention outcomes, and support needed.

The diagnosis of ASD and the scales used for the diagnosis has changed considerably. Changes in the classification and core symptoms also impacted the evaluation scales and, consequently, the reported prevalence rates. ASD, from a rarely diagnosed condition, changed drastically to become a common diagnosis. International prevalence report rates suggest that ASD impacts children between 1 and 2.5% [11,12,13].

Over the past decade, diagnostic services and awareness of ASD have been boosted remarkably in low and middle-income countries (LMICs) [14], and there is reported progress in diagnosis and service provision for less affluent countries [15,16,17,18,19,20]. The consequence of such knowledge boosting is a group of individuals at different age levels classified as at risk of ASD or who have a confirmed diagnosis. Although there is still a long way ahead of the various service providers for ASD in these countries, different scales are available to screen and diagnose individuals with this developmental condition. Most ASD instruments are developed in Western societies and cross-cultural psychometric studies are needed from other cultures/societies [21]. There is a need to demonstrate the applicability of available instruments for ethnic or racial minorities and economically or socially disadvantaged communities, or for those with co-occurring disorders [4].

Additionally, it is essential to understand the degree to which ASD is understood in different communities and the degree to which child development concept is perceived differently in other groups [22]; it is still questioned, considering accumulated evidence on possible cultural/regional differences effects on ASD diagnosis [23]. In addition, to what extent linguistic, construct, and technical equivalence are maintained when ASD instruments are translated, culturally adapted, and psychometrically tested could also be questioned [24].

Regardless of the challenges associated with adopting and normalizing ASD diagnostic scales in LMICs, another challenge is to meet the progress of the screening and diagnostic scales developed in high-income countries. The scales and instruments are generated based on new classifications or in the light of study findings. Some examples are the change in The Autism Diagnostic Observation Schedule (ADOS g to ADOS-2) [25,26], the Childhood Autism Rating Scale (CARS to CARS 2) [27,28], Autism Diagnostic Interview (ADI to ADI-R) [29,30], or different versions of the Gilliam Autism Rating Scale (GARS to GARS-2 and GARS-3) [31,32,33].

Improving the scientific rigor of ASD identification approaches and enhancing accessibility to underserved populations should be prioritized in LMICs [21]. Standardizing a developed scale from one community and understanding its utility in another is complicated. Acquiring knowledge about ASD and the changes in diagnostic criteria make updating the existing scales and developing the new scales a necessary but time and resource-consuming process for the LMICs. The main reason for this challenge is a shortage of infrastructure to update its diagnostic system according to the changes. The consequence is that the standardized older versions of the scales mostly remain applicable regardless of the updated version of the same scale and the changes in the previous data on ASD.

The Gilliam Autism Rating Scale has been updated three times from the mid-90s to the present time. The Gilliam Autism Rating Scale [31], and its subsequent revisions [32,33], is a widely used informant rating instrument for identifying ASD and quantifying its severity in both research and clinical settings. The GARS was developed to cover the identification of the DSM criteria for the disorder and its associated difficulties in children, adolescents, and young adults aged 3–22 years. It was revised as the criteria changed [1,8,9]. The original GARS contains 56 items for measuring stereotyped behaviors, communication and social interaction difficulties, as well as developmental disturbances. Although recommended as a short and effective instrument in determining ASD based on the data from the validation studies [31,32,33,34], it is one of the most used scales globally. Several psychometric studies reported conflicting results for the original version of this scale. While Gilliam [31] reported excellent measurement properties, Lecavalier [35] reported lower levels of reliability and validity, and South et al. [36] and Mazefsky and Oswald [37] reported a low likelihood of correct identification of ASD. Three psychometric studies with the GARS-2 reported more consistent and favorable results. Samadi and McConkey [38], using the Persian version, and Volker et al. [39], with the original, reported good aspects of reliability and validity, such as that the factor structure appeared to be reasonably consistent with the intended constructs of the three GARS-2 subscales [32]. However, their analyses indicated that not all GARS-2 items were assigned to the proposed subscale. Furthermore, sound psychometric data reported for the GARS-3 in the validation study [33] and another study that tested its items across genders [40]; except for the study by Samadi et al. [41], there are no other international peer-reviewed studies to report on its measurement properties. Finally, there is a lack of consistent data to claim the diagnostic test accuracy for all three versions collectively [42].

GARS-3’s application in clinical practice and research settings is recommended commonly. Two significant advantages of using the GARS-3 are that it covers the most recent criteria for the disorder and associated difficulties and takes a dimensional perspective into account. In addition, there are indications that the GARS utilizes family, caregiver, and service provider inputs relevant to culturally and linguistically diverse populations [43]. Nevertheless, cultural/regional effects on its measurements may be reflected through variable levels of sensitivity and specificity across studies through some locally published non-English studies, such as Gorji et al. [44], as reported when using other ASD scales [45], which may result in cross-cultural variability in ASD [2,46].

It is also considered that documenting psychometric properties for one instrument is an iterative process that requires additional studies with different approaches; both second and third versions of GARS need to be further explored in the “at risk of ASD” population to understand the level of reliability of the two versions of the scales in picking up the autistic traits of individuals who have been referred for further evaluation of ASD. Thus, the present study aimed to evaluate the ability of the GARS 2 and 3 to recognize autistic features. Another aim is to understand the level of reliability in a sample of Kurdish individuals who were referred to the Bahoz center for children with developmental disabilities in Erbil, the Kurdistan region of Iraq.

Based on our experience during the last 15 years on ASD in LMICs s, our exploratory investigation focuses on the level of reliability between two versions of the GARS scale on a sample of Kurdish individuals with ASD and ID through an agreement between raters identified cases in two different times.

## 2. Materials and Methods

### 2.1. The Kurdistan Context and Bahoz Center

The Bahoz center is located in the semi-independent Kurdistan region of northern Iraq. The center was established in 2015 by the father of two children with developmental delays. Before launching this center, there were no other professional centers in the region to provide training and rehabilitation services. Although there is no exact result of a formal national census, it is estimated that the population in 2014 was around 5.1 million in nearly 1 million households. The area has a young population; 35% of the population is under 15 years of age, and 13% of the households are estimated to have a member with a type of developmental disability. Children with developmental disabilities do not receive state-provided services such as education, health, and social services; hence, there are governmental fee-paying and private services for this population and their families. Currently, the Kurdistan Region hosts a massive group of refugees that are reported to be around 1.2 million. They are generally displaced Yazidi Kurdish and Syrian Kurdish populations who are displaced due to the conflicts inside Iraq and in neighboring countries.

### 2.2. Participants and Procedure

Data for the present study were derived from the developmental records of individuals presented to the Bahoz center for children with developmental disabilities in Erbil, the Kurdistan region of Iraq, by professionals in the area. The referral aimed to provide training and rehabilitation services. The main inclusion criteria were children and adolescents aged 3–18 years with ASD or other types of developmental disabilities based on psychiatrists, pediatricians, and other professionals in the field of child development. Children with incomplete data from assessments were excluded. Children with ASD or DD were diagnosed following the clinical protocol, which was based on structured interviews with parents, observations of a child, and a complete pediatric and neurological examination report, while the final diagnosis for this purpose was made according to the Diagnostic and Statistical Manual of Mental Disorders [1] (DSM-5; APA, 2013). During the clinical examination, both versions of GARS were administered by trained administrators along with the presence of the individual and his/her parent(s)/caregiver.

The Ethics Committee of the Bahoz center approved the study protocol. The parents/caregivers were informed about the study, and those who agreed to participate and provided written consent were included. The study was conducted under the Declaration of Helsinki (Project identification code = BCRD09-2020 approved on 10 December 2020). In the absence of a clear national protocol, this committee adheres to the seventh revised version of WMA of the Helsinki Declaration on Medical Research involving Human Subjects issued on 19 October 2013.

### 2.3. Detail of the Instruments

The GARS scale was considered the second level of ASD screening [47,48], and in this study, two final versions of the scale have been used.

#### 2.3.1. GARS 2

This scale is divided into different parts; hence, it has only three main subscales that measure a series of behaviors reflecting the three primary areas for the diagnosis of autism based on the DSM-IV-TR criteria for the diagnosis. The subscales are stereotyped behaviors, communication, and social interaction. In addition, an autism index provides a composite indication of autism severity and determines the level of probability of autism’s existence. Respondents must choose from one of the four possible choices provided for each of the 42 Likert-type items. (0 = never observed, 1 = seldom observed, 2 = sometimes observed, and 3 = frequently observed). The developmental section of the scale is answered using an interview with a parent or caregiver who has had sustained contact with the individual from the early stages of their development. The interviewee is asked to answer a series of questions about the child’s development in the first three years of life using yes or no. An autism index of 85 or higher indicates that an individual is likely to have autism. An autism index between 70 to 84 indicates the possibility of having autism, and an autism index score of under 70 indicates that that individual is unlikely to have autism. In the first version, scores of 90 or less indicated that the individual showed a below-average chance of having autism.

#### 2.3.2. GARS 3

The GARS-3 [33] has 58 items measuring specific ASD symptoms and associated difficulties in six subscales: Restricted/Repetitive Behaviors (RRB; 13 items), Social Interaction (SI; 14 items), Social Communication (SC; 9 items), Emotional Responses (ER; 8 items), Cognitive Style (CS; 7 items), and Maladaptive Speech (MS; 7 items). Each item has the same response format ranging from 0 (not at all like the individual) to 3 (very much like the individual). Within each subscale is created a raw score as a sum of all answered items, where higher scores indicate more ASD symptoms/difficulties present. For non-verbal children, raw scores are derived from 44 items from the RRB, SI, SC, and ER subscale, while for verbal children, from all 58 items and the six subscales. Thus, there are four raw scores for an individual who is non-verbal (RB, SI, SC, and ER) and six for a verbal individual (RB, SI, SC, ER, CS, and MS). Using percentile ranks and norms, scaled scores are derived based on each domain’s scores sum, which is used to generate an autism index. An autism index of 54 and lower indicates less likelihood of having ASD, while an autism index between 55 and 70 indicates the probability of ASD and level 1 of ASD, which is based on DSM 5 classification and requires minimal support. An autism index score between 71 and 100 (level 2 and requiring substantial support) and over 101 (level 3 and requiring very significant support) indicates the very likely presence of ASD.

Finally, the updated version of GARS has four new subscales but retained 16 items from its previous version, and 42 new items were added. Additionally, the scoring of both versions is different for verbal and non-verbal children.

To translate and culturally adapt the scales for the Kurdish community, a rigorous process was considered. All 42 items of GARS 2 and 58 items of GARS 3 were deemed to be comprehensive, precise, and relevant for assessing ASD symptoms and emotional, cognitive, and speech difficulties. Thus, none of the items was added, replaced, or omitted. The face and content validity of the translation were found satisfactory since all of the items target ASD symptoms in children and adolescents.

The first author translated both scales from English into Kurdish, considering the usual safeguards of back-translating. The three co-authors reviewed the translated Kurdish versions for language clarity and appropriateness in the Kurdish culture at the first stage of the procedure. All other Bahoz center co-authors were experienced psychologists and therapists.

Both scales were pilot tested with a group of Kurdish caregivers (22 families with different socioeconomic backgrounds for the 3rd version and 15 caregivers for the 2nd version). Collected data indicated that eight of the 58 items (14%) of GARS 3 and five items (12%) out of the 42 items of GARS 2 needed to be reworded to improve their clarity and relevance to the participant’s culture.

The study’s aims were announced to parents and caregivers attending the Bahoz center through the center’s social media pages. The assessors presented information about the project orally before their interview. All the parents and caregivers were seen individually in the Bahoz center, and the individual with ASD was also present at the time of scale administration.

### 2.4. The Scale Administrators

Twelve practitioners undertook the scales in which eight members had a background in psychology (five in clinical psychology, two in special education, and one in educational psychology); four other practitioners also had experience with children with ASD with a background in special education. All practitioners had over two years of experience with children with ASD. The practitioners participated in a three-day workshop on the administration of GARS 3, ASD. Signs and symptoms and ASD criteria in DSM-5. Three of them had three extra training days on the applicability and administration of GARS 2. A certified ADOS and ADI-R trainer provides all the training, administration supervision and in-service training (the first author). In all the scale administration workshops, participants submitted video recordings of interviews undertaken with at least two caregivers of individuals with ASD with the presence of the individual with ASD in the video to improve the level of consistency in administering the scale between the different assessors.

The four evaluators who applied the 2nd version of the scales were excluded from the diagnostic team. The eight members used the 3rd version of the trained group. The time interval between the two diagnoses was under eight weeks.

## 3. Results

It is essential to state that this is an exploratory investigation and statistical analyses (for example, group comparisons, and tests of significance) have only descriptive characteristics of practitioners who have completed GARS 2 and 3 training workshops and worked in the Bahoz center without standardized specialization in ASD.

The Autism Index and the associated levels of ASD are based on normative data from US children. In the present study, data were available for 148 (35 “24%” female and 113 “76%” male) individuals aged from 3 to 19 years of age (Mean = 5.5 SD = 2.6), of which 93 were referred to as individuals with ASD (63%) 55 with other types of DDs, mainly intellectually disabled. All individuals were screened using the Hiva scale, a 10-item screening tool extracted from GARS 2 [49] with a mean score of 3.5 (SD = 2.3). For the ASD group, this rate was higher (Mean = 4.7, SD = 1.7), and for the non-ASD group, the rates were different (Mean = 1.6, SD = 1.6). The correlation between the Hiva screening score and final ASD diagnosis using the 2nd and 3rd versions of the GARS scale is indicated in Table 1.

Ninety-six individuals (65%) scored about the cut-off based on the GARS-2 and 137 individuals (93%) scored above the cut-off based on the GARS-3 (Figure 1).

A similar pattern was seen in two other studies using the second and third versions of GARS in the area; data extracted from these studies and compared with the present study indicated revealed this pattern. Five hundred and ninety-four individuals out of a group of 601 individuals with three types of developmental disabilities using GARS-3 [41] past the cut-off score for ASD. While in another study with GARS-2, 536 children out of a group of 623 children with different types of developmental disabilities past the cut-off score. Figure 2 depicts this pattern in the present and two other previous studies.

The analysis also indicated a significant difference between the screening score for ASD and the non-ASD group, and individuals with ASD had a significantly higher score (X^2^ = 77.1, df = 1, *p* = 0.000). Respondents were 84 couples (57%), while only two respondents (1.4%) were non-parents (siblings). A majority of the sample (64, 43%) were firstborn. One hundred and thirty-nine (89%) individuals were from Erbil city, and the rest were from the other neighboring towns. Sixty-three percent (93 members) became concerned about their children before the second year of their caretakers or earlier.

There was also a significant difference between male and female individuals being diagnosed with GARS-2 (X^2^ = 3.63, df = 1, *p* = 0.046), and female individuals had significantly lower scores in the second version of GARS. At the same time, no similar significance was reported using the third version of the scale (X^2^ = 1.06, df = 1, *p* = 0.244). No correlations were discovered between older and younger individuals with different age cut-offs and versions of GARS.

Out of 93 children who were referred as having ASD, all confirmed the diagnosis using the second version of GARS. While three (5%) out of 55 individuals recruited as having ID also received a diagnosis of ASD (95 cases in all “65%”). Using the 3rd version of GARS increased the level of diagnosis. Not only did all the ASD cases receive the diagnosis, but out of 55 children referred to as having ID, 44 (80%) children received the diagnosis of ASD.

Ninety cases (61%) who received a diagnosis using the 2nd version of the scale received the same diagnosis using the 3rd version. Forty-six cases who did not receive a diagnosis using GARS 2 were diagnosed with GARS-3 (31%). Only six cases (4%) did not receive a diagnosis in the 3rd version of GARS, while receiving a diagnosis using the 2nd version. Only five cases (3%) did not receive a diagnosis using any versions.

The student *t*-test indicated a significant difference between all the non-verbal cases “Autism Index” who scored over cut-off with the 2nd and 3rd versions of the GARS scale (*t* = 3.01, df = 46, *p* = 0.004). At the same time, this rate was not significant for the verbal group (*t* = 0.48, df = 40, *p* = 0.629). For cases referred to as the at-risk of ASD, this difference between the non-verbal subgroup and the verbal group was at a lower level of statistical significance (*t* = 2.53, df = 36, *p* = 0.016) and not significant (*t* = 1.72, df = 19, *p* = 0.101) for the verbal subgroup.

For the non-ASD referred group, the difference between the 2nd and 3rd versions of the scale for the non-verbal sample was close to significant (*t* = 1.84, df = 9, *p* = 0.09), and for the verbal group, it was not significant (*t* = 0.870, df = 20, *p* = 0.395).

Correlations between the diagnosis rate and the version of the used scale were calculated. No significant correlation was seen between samples in two administrations of the 2nd and 3rd versions of the test (*r* = 0.061, *p* = 0.460).

Out of 52 children who, based on the 2nd version, scored under the cut-off score (69), 41 (79%) scored over the cut-off score (55) in the 3rd version and were considered to be at level one of ASD. Figure 3 compared the number of children in each level of diagnosis of ASD (based on the 3rd version) and the probability of ASD (based on the 2nd version of the scale). There was no significant correlation between the age and gender of the diagnosis and version of the scale, and older and younger cases were diagnosed similarly using each scale.

## 4. Discussion

Accessing eligibility to receive ASD services directly depends on the diagnostic criteria; the changing requirements need to be updated or new scales need to be developed for the diagnosis. The changes in the scales might have substantial impacts on service access. Changes to the diagnostic sources such as DSM or ICD cause the diagnostic scales to be updated to meet those changes. The underdiagnosis of ASD cases after the proposed changes to DSM-5 criteria and the scales that became updated based on these changes was a major concern of professionals in the field of developmental disabilities [50]. Hence, data obtained from the utility of the GARS scale updated based on these changes does not support this idea. It was also found that 92.5% of the sample passed the cut-off score for ASD based on the updated version of GARS. Therefore, an increased level of diagnosis was revealed. All the ASD cases reconfirmed the diagnosis along with 44 (80%) out of 55 children who were referred as cases with ID. In contrast, only 5% of the individuals referred to as individuals with ID received an ASD diagnosis based on GARS-2. Application of the GARS-3 with the same group increased this rate to 80%. A possible reason for the higher prevalence rate among individuals with ASD might be the flexibility of the 3rd version in picking up the individuals with comorbid conditions or what is called dual diagnosis. The updated version of the scale is based on the proposed changes in the DSM-5 that permit comorbid diagnoses such as attention-deficit/hyperactivity disorder (ADHD), ID, and ASD simultaneously.

Our results showed that using two scales with a particular group of children yielded different diagnostic reports. It was also predicted that the majority of the ASD cases diagnosed based on the criteria mentioned in the DSM-IV need to receive a similar diagnosis based on the updated version and similar expectations regarding the revised diagnostic scales. Based on the reports [51] and analyzing three extensive databases consisting of a sizable sample out of 5000 individuals who received a diagnosis of ASD (particularly PDD-NOS) based on DSM-IV, it was discovered that 89% to 90% remained eligible for an ASD diagnosis under the proposed DSM-5 [52,53] (Zuddas 2013, Mandy et al., 2012) criteria. Using GARS in this study reported a much higher level of overlap: a rate of 100%. Out of 35.1% (52 children) who scored under the cut-off sore (69) in GARS-2, 79% (41 children) scored over the cut-off sore (55) in GARS-3 and were considered to be at level one of ASD. This was incongruent with the previous prediction that the concern was to exclude cases with higher IQ and those with high-functioning autism or Asperger-like presentations [54].

To understand and examine the number of individuals diagnosed using these criteria and older and updated ASD criteria, some reviews and analyses have been carried out on the sensitivity and specificity of the DSM-IV-(the 2nd version of GARS has been developed based on the ASD criteria presented in this version) and DSM-5 that GARS-3 has been updated based on the changes presented in this version A reduction between 35 to 37% of the individuals who were eligible to be diagnosed as having ASD based on the DSM-5 has been reported [55,56]. Similar findings were reported for the screening scales adopted from the diagnostic scales. The present study showed a significant positive correlation between the Hiva screening scale (adopted from GARS-2), while reporting a lack of significant correlation between the Hiva score and GARS-3. The mixed result of the adopted or translated screening scales for ASD has already been reported in other studies [57]. This finding also indicates the need for updating the screening scales and the diagnostic scales.

Considering findings based on the known group validity results, both versions of the GARS scale scores have sound aspects of this construct validity, since individuals with ASD had significantly higher scores than children with ID [41,49]. Considering that the effect size values ranged from moderate to large, it is indicative that higher scores are more characteristic of severe forms of ASD (levels 3 and 2 based on DSM 5) than those individuals who show milder forms of ASD or ID. In addition, female individuals had significantly lower scores in GARS-2 than male participants. Hence, a similar gender difference was not reported with the 3rd version of the scale. This indicates that expecting higher scores on the Autism Index for male individuals with ASD needs to be investigated and reconsidered. There were irrelevant correlations between both versions of GARS scores and age.

Findings indicated that scales updated based on the DSM-5 tend to diagnose children with less severe symptoms of ASD with older age [58,59]. However, although our finding did not find the difference between the age groups and either scale, picking up individuals with milder forms of symptoms was also observed in our conclusion. Impacts of age should be interpreted with caution, using further monitoring studies with larger sample sizes.

Finally, it seems that professionals and policymakers in low and middle-income countries need to consider the updated versions of the available scales and depend on the older version, which must be considered cautiously.

### Limitation

When interpreting the results of this study, its limitations also need to be considered. The first limitation is that most of our study participants were from one specific center and recruiting samples from the general population may yield different results. Second, the sample size of this study is limited, and this limitation prohibits the applicability of its finding. Third, parents of the recruited individuals with ASD had already become informed about their children’s diagnosis of ASD. This might have impacted their answers to the question in both scales of GARS. Still, we tried to control this impact through the child’s presence at the evaluation session. However, the effects of parental attitudes after the diagnosis still needs to be considered in the analysis. The fourth limitation is that there is no confirmed diagnosis of ASD and ID, or the presence of an accompanying diagnostic condition based on gold standard tests or robust clinical examination in the present study. Although the final diagnosis confirmation of ASD and ID is a desirable factor, in most LMICs, it is a difficult stage to attain due to the scarcity of trained professionals who have the reliability of using the gold standard scales, shortage of resources, and lack of diagnostic tools.

In sum, in the present study’s analysis, except for age and gender, no other factors such as caregivers’ socioeconomic status were considered. In particular, considering clinical variables related to the individuals’ development and ASD manifestation might provide different results.

Nevertheless, the present study is unique because no other studies have compared the reliability of two versions of the GARS scales on a particular group of individuals at two different times. It is also essential that the applicability of both scales has already been tested and approved with this population.

One of the justifications for the present finding might be due to the cultural aspects and different expectations of the cultural groups. A study of ethnic minority groups within one special country, such as the US, has proposed various complexities in the utility of usual scales in diagnosing ASD [60]. Vanegas et al. [61] also reported lower sensitivity and specificity in diagnostic instruments for Latino children with ASD.

Khowaja and colleagues [62] reported that screening scales differ among minority groups, findings that have been reported in other studies [58]. It is also reported that the impacts of changes might be different in various developed countries [60]; therefore, it might be concluded that the application of DSM-5 changes and, consequently, the updated scales in the LMICs need to be understood. Grinker and colleagues [63] suggested that the main challenge of conducting research regarding the utility of the scales is that ASD is viewed differently across cultures and among minority groups. More changes will be presented for the diagnosis of ASD; therefore, updated scales are expected. The Diagnostic and Statistical Manual of Mental Disorders, Fifth Edition, Text Revision (DSM-5-TR), published very recently, included two subtle changes to the criteria for the diagnosis of ASD. Although trivial modifications are unlikely to change diagnostic practice, it makes space for further changes in the future. Thus, classifying DSM-5 for ASD by mapping specific exemplars from evaluation records by a diverse group of clinician raters is feasible and reliable; therefore, the diagnostic scales need to make these changes. This framework provides confidence in the consistency of prevalence classifications of ASD and may be further applied to improve the consistency of ASD diagnoses in clinical settings [64]. Further studies are needed to understand the influence of different factors on the levels of ASD diagnosis among a particular population. In subsequent studies, it would be advisable to be understood if overdiagnosis results from the appropriate sensitivity of the diagnostic tool or its hypersensitivity in this population or even false positive, which is a binary error in screening and diagnostic tools.

## 5. Conclusions

The more analytical changes are presented for ASD by the diagnostic and classification resources, the more updated and tuned scales are needed. It should be helpful to consider this development of the scales in the LMICs and reflect the necessity of using updated versions of the scales for professionals in these countries, regardless of the associated challenges. It is essential to allocate different resources and design several extensive collaborative studies to make the necessary updates in the present applicable diagnostic scales to address this challenge.

## Figures and Tables

**Figure 1 brainsci-12-00537-f001:**
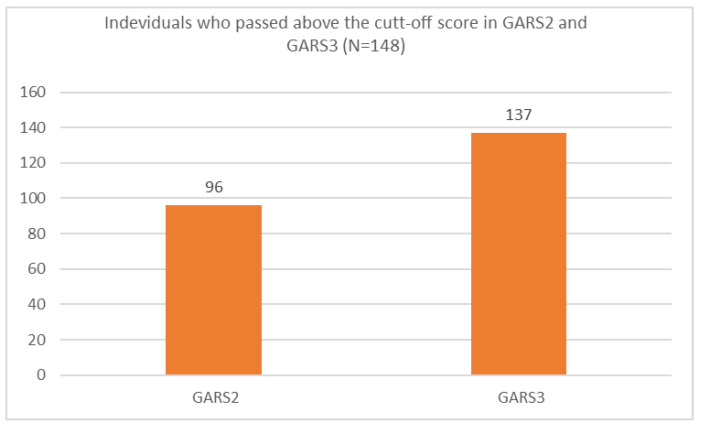
Individuals who passed the cut-off score using GARS-2 and GARS-3 scales in the present study.

**Figure 2 brainsci-12-00537-f002:**
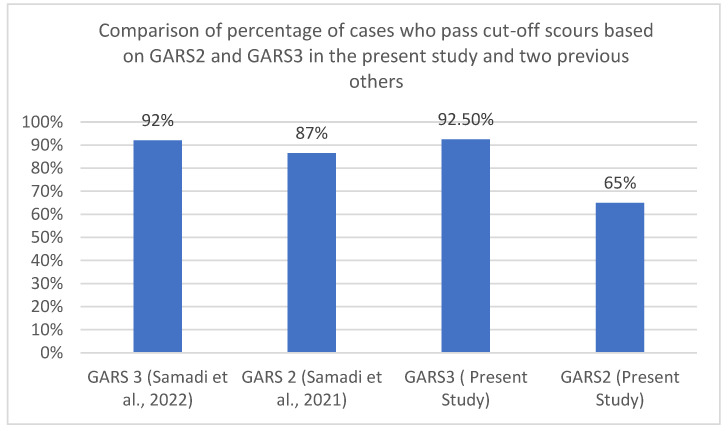
Comparison of the groups who passed the cut-off scours of GARS-2 and GARS-3 in the present study and two previous others.

**Figure 3 brainsci-12-00537-f003:**
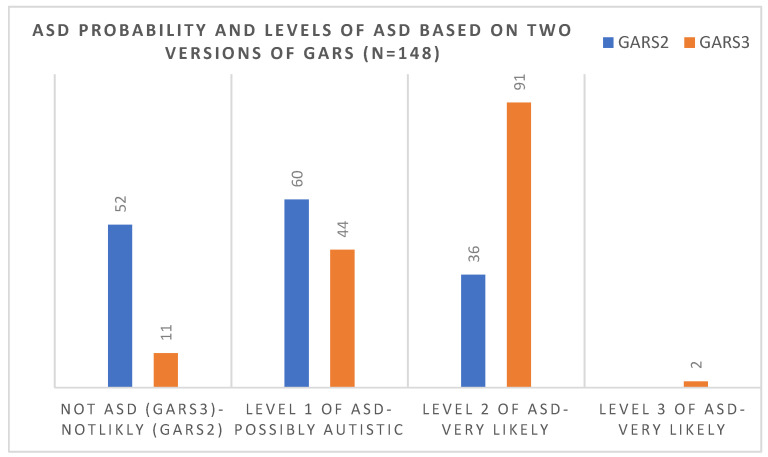
Comparisons of the number of children in each level of diagnosis of ASD (based on the 3rd version) and the probability of ASD (based on the 2nd version of the scale).

**Table 1 brainsci-12-00537-t001:** Correlations of the verbal and nonverbal individuals’ screening score and their final diagnosis in each version of the GARS scale.

	GARS-2 General	GARS-3 General	GARS-2 Non-Verbal	GARS-2 Verbal	GARS-3 Non-Verbal	GARS-3 Verbal
Hiva score	0.547 **	0.073	0.564 **	0.681 **	0.04	0.196

** Correlation is significant at the 0.01 level (2-tailed).

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
