# Peer review of "Autism Spectrum Disorder Diagnostic Criteria Changes and Impacts on the Diagnostic Scales-Utility of the 2nd and 3rd Versions of the Gilliam Autism Rating Scale (GARS)"

_brainsci, 2022, doi:10.3390/brainsci12050537_

Round 1
Reviewer 1 Report
This study evaluates the level of overlap between two different versions of the Gilliam Autism Rating Scale GARS-2 (based on DSM-IV) vs GARS-3 (which is updated based on the DSM-5) on a sample of individuals at risk of having ASD and Intellectual Disability in Iraq. A group of 148 children and teens with ASD and DD was evaluated in order to understand the level of cases that confirm the ASD diagnosis in both scales. Results suggest that the GARS-3 scale, tends to diagnose children with different levels of symptoms severity of ASD at different age.
The topic is very relevant since a correct diagnosis enable the rapid access to rehabilitation service and behavioural intervention.
The paper is fluent and easy to read.
The method is robust. Twelve practitioners over two years of experience with children with ASD, was trained to administered the two versions of the GARS scales. The time interval between the two diagnoses was under eight weeks. Results are statistically analyzed and discussed in details.
Authors also stated the limitations of their study. The sample is small and recruited in a rehabilitation center, so generalization of findings is not possible.
Overall the reader can benefit form this paper.
Minors
Please check English for typos such as:
Abstract - GRAS scale -> GARS scale
Abstract - DD (developmental disabilities)
You first indicate ID subjects and after discuss about DD. Please to clarify.
Author Response
We appreciate the time and kind consideration of reviewer number 1. You can find our feedback on the comments and suggestions in the following:
Reviewer 1:
Comment (s) |
Feedback |
This study evaluates the level of overlap between two different versions of the Gilliam Autism Rating Scale GARS-2 (based on DSM-IV) vs GARS-3 (which is updated based on the DSM-5) on a sample of individuals at risk of having ASD and Intellectual Disability in Iraq. A group of 148 children and teens with ASD and DD was evaluated in order to understand the level of cases that confirm the ASD diagnosis in both scales. Results suggest that the GARS-3 scale, tends to diagnose children with different levels of symptoms severity of ASD at different age. |
|
The topic is very relevant since a correct diagnosis enable the rapid access to rehabilitation service and behavioural intervention |
Thank you for the positive feedback; we agree that a trustworthy and on-time diagnosis will boost the outcomes of the intervention systems. |
The paper is fluent and easy to read. |
We appreciate this very positive feedback |
The method is robust. Twelve practitioners over two years of experience with children with ASD, was trained to administered the two versions of the GARS scales. The time interval between the two diagnoses was under eight weeks. Results are statistically analyzed and discussed in details |
We appreciate the positive comment, and as it is mentioned, we have done our best to adopt the most robust methodology for the study based on our experience.
|
Authors also stated the limitations of their study. The sample is small and recruited in a rehabilitation center, so generalization of findings is not possible.
|
We have mentioned the barriers to the generalization of the finding and also mentioned the strengths of the findings |
Overall the reader can benefit form this paper. |
We appreciate your comment |
Please check English for typos such as: Abstract - GRAS scale -> GARS scale Abstract - DD (developmental disabilities) You first indicate ID subjects and after discuss about DD. Please to clarify.
|
All the typos were corrected, and DD at the discussion part changed to ID. |

Reviewer 2 Report
The presented work in a factual and condensed way presents an important problem concerning diagnostic procedures in developing countries.
The presented research is exploratory in nature, but it does not diminish its importance in any way. The presented problem actually concerns not only developing countries, but also Central and Eastern European countries, where funding for the development of new test versions is limited.
The questions and hypotheses posed are extremely important in the context of the rapid development of the psychological testing market and the changes taking place in the new classifications of mental diseases and disorders. Failure to keep up with changes may result in insufficient precision in measuring diseases and mental disorders in certain areas of the world. In addition, authorities may not see the importance of implementing newer versions of diagnostic tools without having data on the importance of this procedure based on national data.
The presented work fills a very important gap in the knowledge of culturally dependent diagnostic procedures. Despite the assumption that the symptoms of disorders should be universal in nature, the presented work makes us pay attention to cultural factors in the diagnostic process.
The authors have carefully analyzed the limitations of the study and report them clearly and reliably. However, it should be acknowledged that the specificity of the research and the policy of the country in which the research was carried out somewhat limit the possibility of avoiding these limitations. Despite the obvious limitations and the need for further research, the work is valuable and allows us to draw attention to the important problem of contemporary psychology, which is Western-centrism. It is necessary to conduct research and to publish its results more widely in developing countries, with the assumption that initially they will be mainly exploratory in nature.
In subsequent studies, it would be advisable to consider whether the percentage of children with autism who were identified in the group of children with intellectual disability results from the appropriate sensitivity of the diagnostic tool or its hypersensitivity in this population. Are we not struggling with the situation of false positive indicators? In addition, it would be interesting for the authors to include information in the description of the study group whether there were also children with intellectual disabilities (i.e. dual diagnosis) in the group of children with autism. It would be advisable to describe the characteristics of the study group more precisely, especially the group with autism, and to indicate what percentage of children received, in addition to autism, an additional diagnosis.
Author Response
We appreciate the time and kind consideration of reviewer number 2. You can find our feedback on the comments and suggestions in the following:
Comment (s) |
Feedback |
The presented work in a factual and condensed way presents an important problem concerning diagnostic procedures in developing countries. |
Thank you for the comment, and we are humbled to know that we could draw attention to an important but neglected aspect of the diagnosis DDs in LMICs. |
The presented research is exploratory in nature, but it does not diminish its importance in any way. The presented problem actually concerns not only developing countries, but also Central and Eastern European countries, where funding for the development of new test versions is limited. |
Thank you for the very positive comment, and it is our privilege to notice the broader applicability of our presented findings in this study. |
The questions and hypotheses posed are extremely important in the context of the rapid development of the psychological testing market and the changes taking place in the new classifications of mental diseases and disorders. Failure to keep up with changes may result in insufficient precision in measuring diseases and mental disorders in certain areas of the world. In addition, authorities may not see the importance of implementing newer versions of diagnostic tools without having data on the importance of this procedure based on national data. |
We totally agree with this justification and as the researchers in the field of DDs in LMICs witness the impacts of this rapid updating and changes in the area of mental health support, particularly in communities with the absence of the infrastructure and resources |
The presented work fills a very important gap in the knowledge of culturally dependent diagnostic procedures. |
We appreciate the comment |
Despite the assumption that the symptoms of disorders should be universal in nature, the presented work makes us pay attention to cultural factors in the diagnostic process. |
That is our main idea for the study and the main aim in our attempts for publication. |
The authors have carefully analyzed the limitations of the study and report them clearly and reliably. However, it should be acknowledged that the specificity of the research and the policy of the country in which the research was carried out somewhat limit the possibility of avoiding these limitations. |
We appreciate the comment and the level of your awareness regarding the imposed limitations due to legislative and cultural barriers. |
Despite the obvious limitations and the need for further research, the work is valuable and allows us to draw attention to the important problem of contemporary psychology, which is Western-centrism. |
We appreciate your comment and wholeheartedly agree with the justification |
It is necessary to conduct research and to publish its results more widely in developing countries, with the assumption that initially, they will be mainly exploratory in nature. |
We agree and express our sincere agreement with participation in any study projects that increase the level of awareness regarding DDs such as ASD in LMICs |
In subsequent studies, it would be advisable to consider whether the percentage of children with autism who were identified in the group of children with intellectual disability results from the appropriate sensitivity of the diagnostic tool or its hypersensitivity in this population. Are we not struggling with the situation of false positive indicators? |
We have added a part of this suggestion to the further studies paragraph at the end of the discussion, lines 560 to 466: Further studies are needed to understand the influence of different factors on the levels of ASD diagnosis among a particular population. In subsequent studies, it would be advisable to be understood if overdiagnosis results from the appropriate sensitivity of the diagnostic tool or its hypersensitivity in this population or even false positive, which is a binary error in screening and diagnostic tools. |
In addition, it would be interesting for the authors to include information in the description of the study group whether there were also children with intellectual disabilities (i.e. dual diagnosis) in the group of children with autism. It would be advisable to describe the characteristics of the study group more precisely, especially the group with autism, and to indicate what percentage of children received, in addition to autism, an additional diagnosis. |
We agree with the importance of the requested information regarding the presence of accompanying diagnostic conditions. Since no information was available, we added the following information to the limitation part:
The fourth limitation is that there is no confirmed diagnosis of ASD and ID or presence of an accompanying diagnostic condition based on the gold standard tests or robust clinical examination in the present study. |
